# Fast and efficient purification of SARS-CoV-2 RNA dependent RNA polymerase complex expressed in *Escherichia coli*

Clément Madru[1], Ayten Dizkirici Tekpinar[1,2], Sandrine Rosario[1], Dariusz Czernecki[1,3], Sébastien Brûlé[4], Ludovic Sauguet[1]*, Marc Delarue[1]*

**1** Unit of Structural Dynamics of Macromolecules, Institut Pasteur & CNRS UMR, Paris, France, **2** Department of Molecular Biology and Genetics, Faculty of Science, Van Yüzüncü Yıl University, Van, Turkey, **3** École Doctorale Complexité du Vivant, Sorbonne Université, Paris, France, **4** Molecular Biophysics Platform, C2RT, Institut Pasteur, CNRS UMR, Paris, France

☯ These authors contributed equally to this work.
* ludovic.sauguet@pasteur.fr (LS); marc.delarue@pasteur.fr (MD)

**Data Availability Statement:** All relevant data are within the paper and its Supporting Information files.

## Abstract

To stop the COVID-19 pandemic due to the Severe Acute Respiratory Syndrome Coronavirus 2 (SARS-CoV-2), which caused more than 2.5 million deaths to date, new antiviral molecules are urgently needed. The replication of SARS-CoV-2 requires the RNA-dependent RNA polymerase (RdRp), making RdRp an excellent target for antiviral agents. RdRp is a multi-subunit complex composed of 3 viral proteins named nsp7, nsp8 and nsp12 that ensure the ~30 kb RNA genome's transcription and replication. The main strategies employed so far for the overproduction of RdRp consist of expressing and purifying the three subunits separately before assembling the complex *in vitro*. However, nsp12 shows limited solubility in bacterial expression systems and is often produced in insect cells. Here, we describe an alternative strategy to co-express the full SARS-CoV-2 RdRp in *E. coli*, using a single plasmid. Characterization of the purified recombinant SARS-CoV-2 RdRp shows that it forms a complex with the expected (nsp7)(nsp8)$_2$(nsp12) stoichiometry. RNA polymerization activity was measured using primer-extension assays showing that the purified enzyme is functional. The purification protocol can be achieved in one single day, surpassing in speed all other published protocols. Our construct is ideally suited for screening RdRp and its variants against very large chemical compounds libraries and has been made available to the scientific community through the Addgene plasmid depository (Addgene ID: 165451).

## Introduction

The COVID-19 pandemic caused by the Severe Acute Respiratory Syndrome Coronavirus 2 (SARS-CoV-2) has affected millions of people with a death toll exceeding two million worldwide [1–4]. SARS-CoV-2 is an enveloped, single-stranded virus with a positive-sense RNA genome [5, 6]. One of the most promising druggable targets is the RNA-dependent RNA polymerase (RdRp), a central element of SARS-CoV-2 life cycle, responsible for the transcription

**Funding:** This work was funded by the Institut Pasteur, and by an ANR JCJC grant ANR-17-CE11-0005-01. The post-doctoral fellowship of C.M is funded by the Pasteur-Roux-Cantarini fellowship from the Institut Pasteur. The fellowship of D.C was funded by Sorbonne University ED515.

**Competing interests:** The authors have declared that no competing interests exist.

and replication of the ~30kb genome [7–10]. RdRp from coronaviruses are error-prone enzymes [11], recognizing various modified nucleosides analogs (NAs) as substrate as well. Such NAs may disrupt viral RNA synthesis via chain termination, making them important candidates as anti-viral agents [12–14]. Early efforts to discover treatments were focused on evaluating the efficacy of already known NAs against SARS-CoV-2. Among them, Remdesivir was initially approved to treat Ebola [15] and received U.S. Food and Drug Administration (FDA) approval for COVID-19 treatment, while Favipiravir, currently licensed in Japan for use in the treatment of influenza virus [16], is being evaluated in clinical trials [17, 18]. Considering the global impact of the COVID-19 pandemic, the new variants appearance and the possibility of re-emergence of coronavirus infections in the future, there is an urgent need to develop new antiviral agents specifically targeting the RdRp involved in pivotal steps of SARS-CoV-2 pathogenesis.

The production of sufficient amounts of heterologous RdRp with a native structure and full biological activity is a prerequisite for the discovery, optimization and comprehensive evaluation of new drugs directed against SARS-CoV-2, including in High Throughput Screening (HTS) assays involving huge chemical libraries. RdRp is composed of 3 viral non-structural proteins (nsp) named nsp7, nsp8 and nsp12. The core component nsp12 hosts the catalytic polymerase activity, greatly enhanced by the two accessory subunits nsp7 and nsp8 [19, 20]. RdRp has been a subject of intensive structural biology efforts, yielding high resolutions cryo-EM structures of the RdRp in its apo form [21, 22], or bound with RNA [23–25], inhibitors [24–27], and to other factors [28–30]. These structures showed the nsp12 core bound to a heterodimer nsp7-nsp8 and an additional nsp8 at a different binding site. The two nsp8 copies expose long N-terminal alpha helices that slide along the exiting RNA to prevent premature dissociation [23]. The production of intact and correctly assembled (nsp7)(nsp8)$_2$(nsp12) complexes is therefore required to get fully active and processive polymerase.

The main strategies employed so far for the overexpression of recombinant RdRp consists in expressing and purifying the catalytic nsp12 subunit and the accessory nsp7-nsp8 subunits separately before assembling the complex *in vitro* [11, 22, 24, 28–30]. However, while nsp7 and nsp8 express readily in *Escherichia coli*, nsp12 shows limited solubility in bacterial expression systems and is often produced in insect cells [22, 23, 25–27]. These approaches multiply the protein expression and purification steps, making RdRp isolation cost- and time-consuming. Here, we describe an alternative strategy to co-express the SARS-CoV-2 RdRp directly in *E. coli*, using a single plasmid. Characterization of the purified recombinant SARS-CoV-2 RdRp using analytical ultracentrifugation assays (AUC) revealed that it forms an heterotetramer with the expected stoichiometry. RNA polymerization activity was measured using primer-extension assays and showed that the purified enzyme is indeed functional. This approach provides a useful alternative to more expensive and complicated protein expression systems, and offers many practical advantages inherent to bacterial systems, such as easy generation of mutants and simple cultivation handling. Our fast, single-day purification protocol results in a stable and active complex that can be used in most protein biochemistry laboratories for drug screening as well as for functional studies.

## Materials and methods

### Optimization of recombinant RdRp expression

**Cloning.**  The open reading frames (ORF) of the nsp7, nsp8 and nsp12 genes from SARS-CoV-2 virus (GenBank: MN908947.3) were optimized for expression in *E. coli*, synthesized commercially by geneArt (Thermo Fisher) and inserted into a modified pRSFDuet-1

(Novagen), resulting in pRSFDuet-1(14his-nsp8/nsp7)(nsp12) and the pRSFDuet-1(14his-nsp12)(nsp7/nsp8) (S1 and S2 Figs in S1 File).

**Optimal inducer concentration.** BL21 Star (DE3) strain from *E. coli* (Thermo Fisher) containing pRSFDuet-1(14His-nsp8/nsp7)(nsp12) was grown overnight in Lysogeny broth medium supplemented with 100 μg/mL kanamycin (LBK). A fresh culture was then inoculated (1:100) and incubated at 37°C, 180 rpm. When its optical density at 600 nm ($OD_{600}$) reached 0.6, the culture was divided into 2 series of 6x10 mL in 50 mL tubes, and the induction was made individually with IPTG concentrations of 0, 0.05, 0.1, 0.2, 0.5 and 1 mM. After 3 hours at 37°C and 20 hours at 20°C, $OD_{600}$ was measured and 1 mL aliquots were centrifuged. Cells were resuspended in 1X Lithium dodecyl sulfate (LDS) sample buffer (Invitrogen) supplemented with 10 mM dithiothreitol (DTT) at a concentration of 10 $UOD_{600}$/mL. Samples were then vortexed for 30 sec, boiled for 5 min and centrifuged for 2 min at 20 000 g. 5 μL of supernatant were finally loaded on SDS-PAGE 4–20%.

**Optimal post-induction temperature and post-incubation time.** A culture of BL21-Star-(DE3)-pRSFDuet-1(14his-nsp8/nsp7)(nsp12) was grown at 37°C, 180 rpm, and split into 3x100 mL in 500 mL flasks at $OD_{600}$ 0.6. The recombinant protein expression was then induced by adding 0.05 mM IPTG at 20°C, 30°C and 37°C, 180rpm. $OD_{600}$ were measured and 1 mL aliquots were taken after 2, 4, 6, 8 and 20 hours. Loading samples for SDS-PAGE analysis were prepared as described above.

## Large scale production of recombinant RdRb

100 ng of pRSFDuet-1(14-his-nsp8/nsp7)(nsp12) were added to 50 μL chemically competent BL21 Star (DE3) and incubated for 30 min in ice. Cells were heat-shocked for 30 sec at 42°C and incubated in ice for 2 min. 900 μL of super optimal broth (SOC) medium were added and the mixture incubated at 37°C for 1 h. 200 mL of LBK were then inoculated with the transformation reaction in 1 L flask, and cells were grown at 37°C overnight with 180 rpm agitation. 30 mL of the overnight culture were inoculated (1:100) into 1 L of LBK in 5 L flasks and incubated at 37°C with 180 rpm agitation for a few hours until $OD_{600}$ reaches 0.6. Flasks were then placed for 15 min at 4°C and recombinant protein expression was induced by adding 0.05 mM isopropyl-β-D-1-thiogalactopyranoside (IPTG) and incubating 20 hours at 20°C with 180 rpm agitation. Cells were harvested by centrifugation, washed once with fresh LB, and stored at -80°C.

## Purification of recombinant RdRp

Cells were resuspended in the HisTrap buffer A (50 mM Na-HEPES at pH 8, 500 mM NaCl, 10 mM imidazole) supplemented with complete EDTA-free protease inhibitors (Thermo Fisher) and 500 units of benzonase (Sigma) at 4°C. Resuspended cells were then lysed by mechanical disruption with 3 passes through a pre-cooled cell disruptor (Constant System Limited) at 1.4 kPa, and the lysate was centrifuged at 20 000 g for 30 min at 4°C. All the following steps described below were performed with chromatography columns from GE Healthcare connected to an ÄKTA pure system (GE Healthcare) at 4°C. After centrifugation, the clear supernatant containing the RdRp complex was loaded onto a 5 mL HisTrap nickel affinity column (GE Healthcare). The column was then washed with 25 mL of 5% HisTrap buffer B (50 mM Na-HEPES pH 8, 500 mM NaCl, 500 mM imidazole). The complex was finally eluted using a 50 mL linear gradient of imidazole (5%-100% HisTrap buffer B). Fractions were analyzed by SDS-PAGE 4–20%, and DNA contamination was detected by measuring the ultraviolet (UV) absorption spectra and the ratio of absorbance at 260 nm vs 280 nm (A260/280). RdRp-containing HisTrap fractions were combined and 5-fold diluted in a 50mM Na-HEPES

pH 8 solution before being loaded onto a 5 ml HiTrap Q HP anion exchange column (GE Healthcare), pre-equilibrated in the HiTrap Q buffer A (50 mM Na-HEPES pH 8, 150 mM NaCl). The column was washed with 25 mL of HiTrap Q buffer A while protein complex was eluted with a 50 mL linear gradient of NaCl realized by mixing HiTrap Q buffer A with HiTrap Q buffer B (50 mM Na-HEPES pH 8, 1 M NaCl). The purest fractions containing RdRp complex were combined and concentrated up-to 5 mg/mL using Amicon Ultra-4 centrifugal filter units 30 000 NMWL (EMD Millipore). The purification final step involved a size-exclusion chromatography on a Superdex 200 10/300 equilibrated in S200 buffer (20 mM Na-HEPES pH 8, 300 mM NaCl, 1 mM MgCl2). RdRp containing fractions were then combined, concentrated up-to 3 mg/mL, flash-frozen in liquid nitrogen and stored at -80˚C.

### Analytical ultracentrifugation assays

Sedimentation velocity experiment was performed with a Beckman Coulter Optima analytical ultracentrifuge (Beckman-Coulter, USA) with an An-60 Ti rotor at 20˚C. The freshly purified RdRp complex at a concentration of 2.7 mg/ml was centrifuged at 35,000 rpm in 3-mm double-sector epoxy centerpieces. 100 scans were collected at 1 min intervals with a radial step size of 0.001cm. Detection of the protein complex as a function of radial position and time was performed by absorbance measurements at 250 nm, 280 nm and by interference detection. Profiles were analyzed using the continuous (s) distribution model of the software Sedfit [31]. The partial specific volume of the protein, of 0.733 was theoretically calculated in Sedfit. The buffer viscosity of 0.01046 Poise and the buffer density of 1.0128 were respectively determined with the Viscosizer TD (Malvern Panalytical, UK) and the DMA 5000 M (Anton Paar).

### RNA primer extension assays

The RNA duplex was prepared by mixing a 40-mer RNA template (5'-CUAUCCCCAUGUGA UUUUAAUAGCUUCUUAGGAGAAUGAC-3') corresponding to the 3' end of SARS-CoV-2 genome with a 20-mer fluorescent RNA primer (5'-FAM-GUCAUUCUCCUAAGAAGCUA-3') in water. The mix was then heated for 2 min to 70 ˚C and slowly cooled to room temperature. Primer extension assays were performed with 500 nM of purified RdRp complex in the presence of 50 μM NTPs and 50 nM RNA duplex, in a reaction buffer containing 20 mM Na-HEPES pH 8, 50 mM NaCl, and 3 mM MgCl2. The reaction was carried out in 150 μL at 37˚C. After 30 sec, 1, 5, 10, 15, and 30 min, 10 μL were pipetted from the reaction mix and immediately diluted in 20 μL of formamide gel-loading buffer (10 mM EDTA, 0.1 mg/mL Xylene Cyanol, 0.1 mg/mL bromophenol blue, 95% formamide). Reactions were then stopped by boiling samples for 3 min at 90˚C. 5 μL samples were loaded in 40 x 30 cm urea-polyacrylamide (8%) gel and separated for 3 hours at 1500V. Images were taken using a typhoon FLA 9000 (GE Healthcare). For the primer extension assay in presence of RdRp eluted from nickel affinity chromatography, the concentration of the native (nsp12)(nsp8)$_2$(nsp7) could not be accurately determined due to an excess of nsp7 and nsp8. The fraction used for the activity assay showed an $A_{280}$ of 8.0 after 1 hour of dialysis in S200 buffer.

## Results and discussion

### Optimization of recombinant RdRp expression

**Recombinant vector construction.** The nsp7, nsp8 and nsp12 genes were optimized for recombinant expression in *E. coli*, and inserted into a modified pRSFDuet-1 vector, allowing the expression of 14 histidine (14his) N-fused proteins. This vector encodes two multiple cloning sites (MCS1 and 2), each of which is preceded by a T7 promoter, lac operator, and

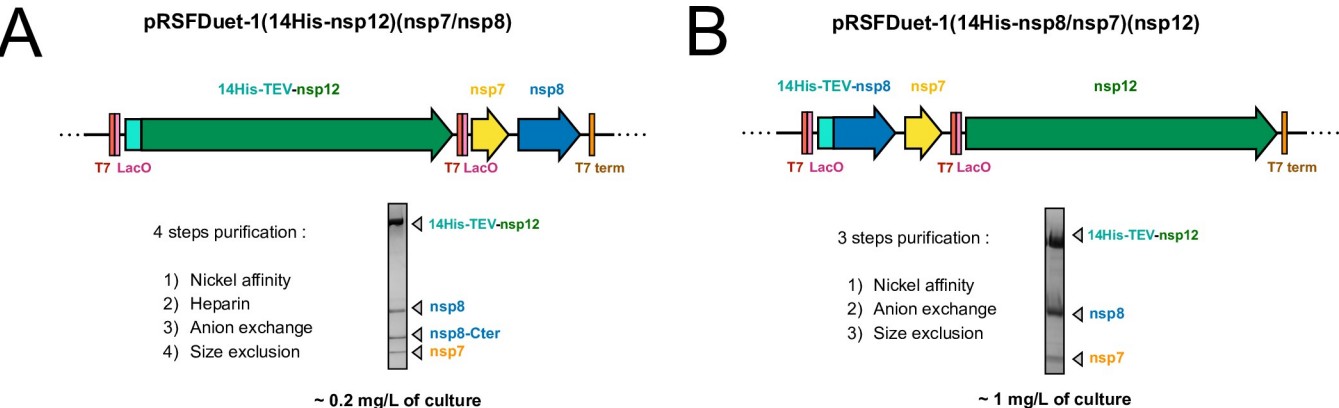

**Fig 1. Recombinant expression vectors.** Top: ORF arrangement in pRSFDuet-1(14his-nsp8/nsp7)(nsp12) (A) and pRSFDuet-1(14his-nsp8/nsp7)(nsp12) (B) employed in this study. Nsp7, nsp8 and nsp12 ORFs were cloned in a modified pRSFDuet-1, allowing the expression of TEV-cleavable 14xhistidine (14his-TEV) N-fused protein. The vector encodes two multiple cloning sites (MCS1 and MCS2) each of which is preceded by a T7 promoter (T7), a lac operator (LacO) and a ribosome binding site. The fully annotated plasmid maps are given in S1 Fig in S1 File. Below: representative analysis by SDS-PAGE (4–20%) of the purified RdRp, purification pathway and yields are given for each plasmid.

ribosome binding site (RBS). Two different cloning strategies were employed to express the RdRp complex, giving two plasmids named pRSFDuet-1(14his-nsp12)(nsp7/nsp8) and pRSFDuet-1(14his-nsp8/nsp7)(nsp12) (Fig 1A). The pRSFDuet-1(14his-nsp12)(nsp7/nsp8) was first evaluated to achieve the production of recombinant RdRp. Early expression and purification trials yielded ~0.2 mg of recombinant RdRp per liter of culture. The purified complex was functionally active but partially proteolyzed, exhibiting a fourth band on SDS-PAGE analysis after four purification steps (S3 Fig in S1 File). This additional band corresponding to an approximately 12 kDa protein was identified as the globular C-terminal part of nsp8, showing that the N-terminal region of nsp8 (~80 amino-acids) is degraded during the purification process. This flexible region becomes ordered when the RNA duplex exits from the enzyme's active site, making extensive interactions with the RNA molecule [23]. This proteolysis could not be suppressed despite all our efforts and optimizations. Therefore, we designed another construct, the pRSFDuet-1(14his-nsp8/nsp7)(nsp12), which was tested for RdRp production and showed promising results (Fig 1B). Indeed, only the three expected bands nsp7, nsp8 and nsp12 are observed in SDS-PAGE analysis from the first affinity purification step. Besides avoiding the purification of proteolyzed nsp8 by fusing the tag to the sensitive region, this specific arrangement also leads to an increased production of the catalytic subunit nsp12, often limited in bacterial system. Indeed, the mRNA transcripts of the second MSC2 can be generated independently of the "read-through" transcript from the first promoter. Purification assays using the pRSFDuet-1(14his-nsp8/nsp7)(nsp12) construct yield ~1 mg of pure full-length complex per liter of culture, thereby allowing a markedly improved RdRp production compared to the pRSFDuet-1(14His-nsp12)(nsp7/nsp8) construct.

**Optimal inducer concentration, induction temperature and post-induction incubation time.** The yield of BL21-Star-(DE3)-pRSFDuet-1(14his-nsp8/nsp7)(nsp12) cells was optimized for IPTG concentration, incubation time and temperature. First, the influence of inducer concentration was monitored by adding IPTG at $OD_{600}$ 0.6 to the final concentrations of 0.05, 0.1, 0.2, 0.5 and 1 mM. As depicted in Fig 2A, varying IPTG concentration did not significantly affect the recombinant expression level at 20°C for 20 hours and 37°C for 3 h. It was however set to the lowest value 0.05 mM, because of its negative effect on cell growth.

Effects of post-induction temperature and post-induction incubation time on RdRp production were also investigated. After induction with 0.05 mM IPTG, cells were incubated at

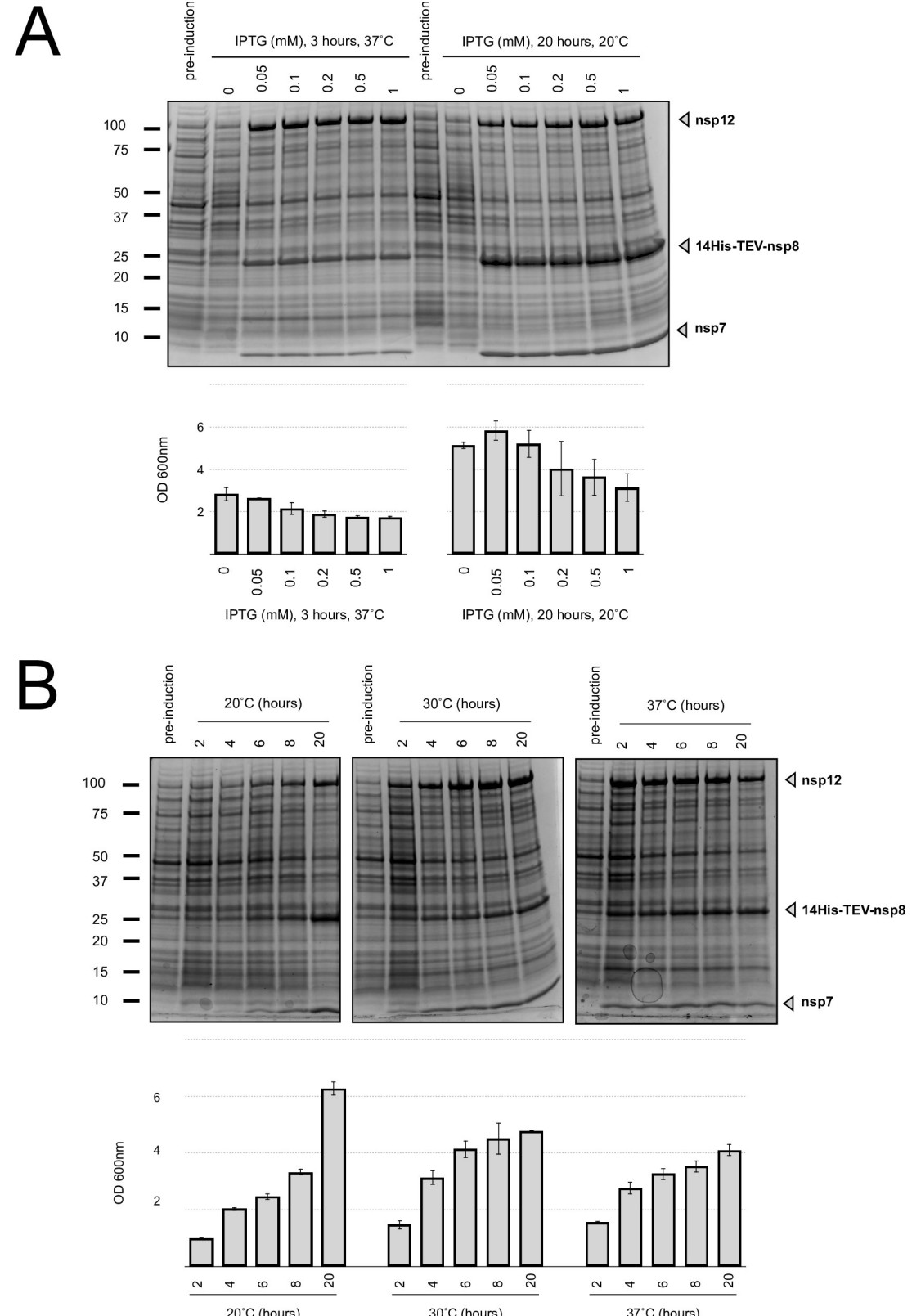

**Fig 2. Optimization of the SARS-CoV-2 RdRp recombinant expression in E. *coli*.** (A) RdRp recombinant expression levels and cell growth at various IPTG concentrations. IPTG concentrations of 0.05, 0.1, 0.2, 0.5, and 1 mM were added to the culture

in the mid-exponential growth phase. After 3 hours incubation at 37˚C (left) or 20 hours at 20˚C (right), the total cell extracts were analyzed on SDS-PAGE (4–20%) and the cell biomass productions were determined. Error bars represent 1 SD (n = 3). (B) Comparison of RdRp production and cell growth at various post-induction incubation times and temperatures. Total cell lysates were analyzed on SDS PAGE (4–20%) and final biomass were determined after 2, 4, 6, 8 and 20 hours at 20˚C, 30˚C and 37˚C. Error bars represent 1 SD (n = 3).

20˚C, 30˚C and 37˚C for 2, 4, 6, 8 and 20 hours and lysed; the lysates were analyzed on SDS-PAGE (Fig 2B). 20 hours incubations at 20˚C and 30˚C have given highest ratios of recombinant RdRp in cell lysate per liter of culture. In order to choose the best incubation temperature, purification of recombinant RdRp from 20˚C and 30˚C *E. coli* cultures were performed using the same protocol (see below). While the final yields were similar, the complex purified from the 20˚C-incubated cells appeared purer, with fewer host contaminants in eluted fractions from the first purification step (S4 Fig in S1 File).

**Purification of recombinant RdRp.** The purification protocol consists of three successive chromatographic steps, including nickel affinity, anion exchange, and size exclusion columns. Nsp8 is fused to an N-terminal 14his-tag, enabling large-scale purification of the recombinant RdRp using nickel affinity chromatography. As shown in Fig 3A, weakly bound proteins such as host proteins were mostly washed out using a low concentration of imidazole (30 mM), while the eluted fractions (75–400 mM) showed 3 bands on SDS-PAGE, at the range of 105 kDa, 25 kDa and 10 kDa, corresponding respectively to nsp12, 14his-nsp8 and nsp7. Relative band quantification revealed a large excess of the 14his-nsp8 and nsp7 accessory subunits compared to nsp12. An HiTrap Q ion exchange chromatography is thereby needed to remove the accessory subunits excess. The elution profile exhibits a unique peak containing the RdRp complex while small subunits alone flow through the column at pH 8 in 150 mM NaCl (Fig 3B). The final step of purification uses a size-exclusion chromatography step that results in one single peak (Fig 3C). An overall yield of 1.0 ± 0.2 mg of complex is obtained per liter of *E. coli* culture. The UV spectrum of the purified complex did not reveal DNA contamination with a A260/A280 ratio of 0.6. Depending on the applications, the N-terminal 14his-tag fused to nsp8 can be removed following TEV-protease cleavage. We recommend to do so after the HiTrapQ step. The RdRp could thus be incubated overnight with TEV-protease at 4˚C in the presence of 1mM DTT before the final size exclusion chromatography. However, we did not verify RdRp activity after cleavage in the present study. While this manuscript was under review, a method for producing tag-free SARS-CoV-2 RdRp in *E. coli* has been reported by Dangefield *et al.* [32]. Their approach requires many more steps than traditional purification of tagged proteins, but yields 7 mg of active enzyme per liter of culture.

**Biophysical and functional characterization of the purified RdRp complex.** Analytical ultracentrifugation revealed that the purified RdRp is homogeneous, exhibiting a main peak at 280 nm, with a sedimentation coefficient of 6.2 S. The combination of the sedimentation coefficient and the integration of absoprtion at 250 nm, 280 nm as well as the interference signal allows to resolve the complex stoichiometry using our in-house protocols. The molecular mass of 160 kDa displayed by Sedfit and the peak integration analysis are fully compatible with the expected $(nsp7)(nsp8)_2(nsp12)$ stoichiometry with a theoretical molecular mass of 169 kDa including the two flexible 14His tags (4 kDa). Also, the frictional ratio of 1.55 suggests that the heterotetramer is slightly elongated (Fig 4A).

In addition, an RNA primer extension assay was performed to verify whether the purified RdRp was active. The purified RdRp was incubated for 1 hour with a 40-mer RNA template mimicking the 3'-extremity of the viral genome, primed by a fluorescently-labeled 20-mer RNA. As shown in Fig 4B, the purified RdRp is active and mediates primer-dependent RNA elongation. Moreover, both fresh and thawed protein samples showed similar activity,

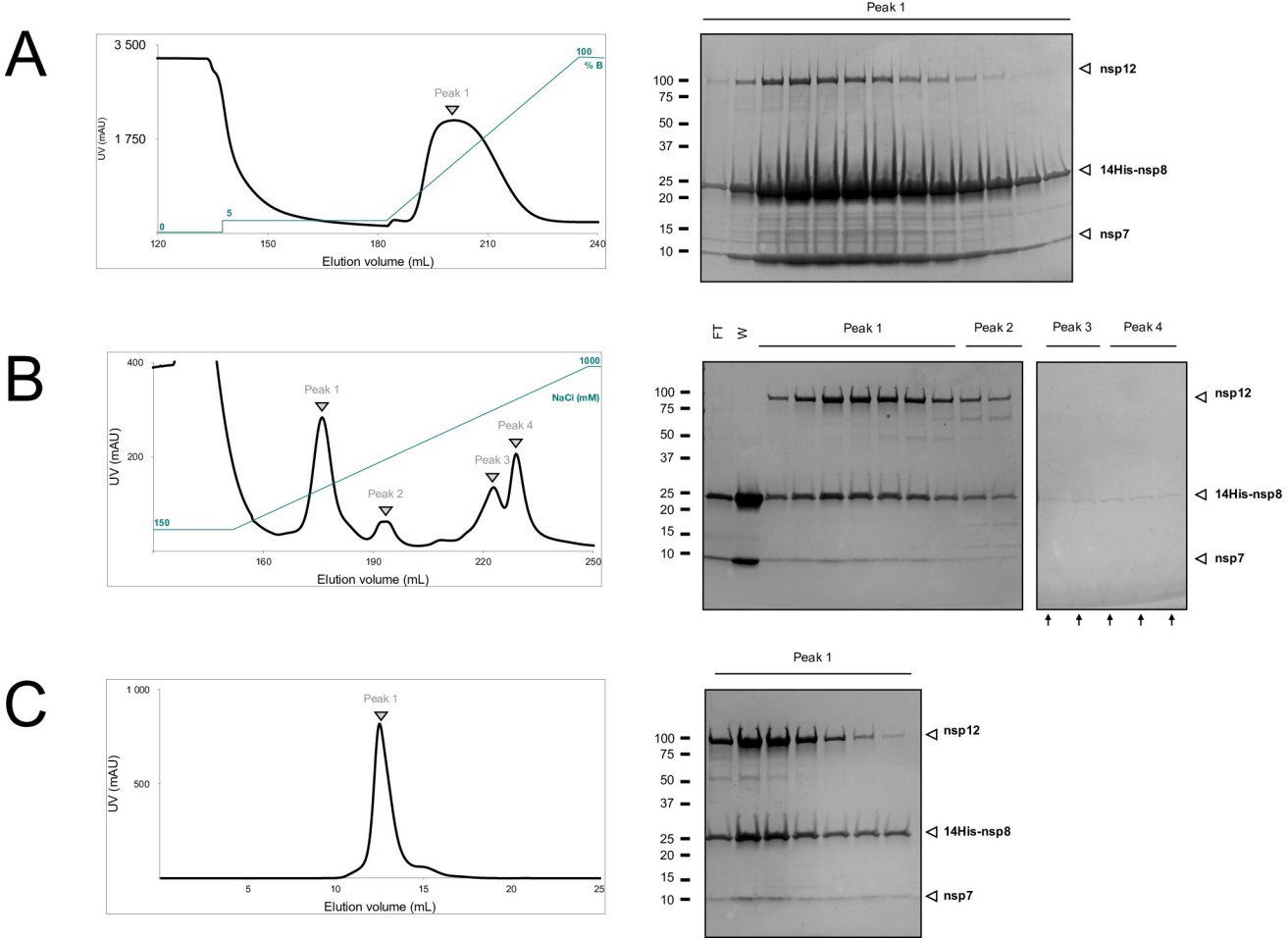

**Fig 3. Purification of the recombinant SARS-CoV-2 RdRp complex.** Representative elution profiles and associated SDS-PAGE (4–20%) analysis from each purification step. **(A)** Nickel affinity chromatography. Elution was performed with an imidazole gradient (10–500 mM). **(B)** Anion exchange chromatography. Elution was performed with an NaCl gradient (150–1000 mM). Fractions containing nucleic acids are indicated with black arrows. **(C)** Size exclusion chromatography in 20 mM Na-HEPES pH 8, 300 mM NaCl, 1 mM MgCl2.

suggesting that freezing does not affect enzyme activity (Fig 4C). Finally, we showed that the RdRp eluted from the nickel pseudo-affinity chromatography is already active (Fig 4C). Albeit containing an excess of the nsp7 and nsp8 subunits, active RdRP can be obtained from one single purification step. In particular, this approach offers many practical advantages for generating and screening RdRp variants. This could facilitate the first steps of the high throughput mutant characterization by shortening the purification time at the cost of sample purity.

In summary, the quality of the purified RdRp has been fully assessed using biophysical and biochemical assays. The sample has the expected stoichiometry, is homogeneous and is functionally active.

## Conclusion

Motivated by *E. coli*'s broad accessibility, ease of culture, rapid growth rates and proven scalability, we developed an efficient expression and purification system for the SARS-CoV-2 RdRp complex in this bacterial host. Active RdRp can be immobilized and isolated from

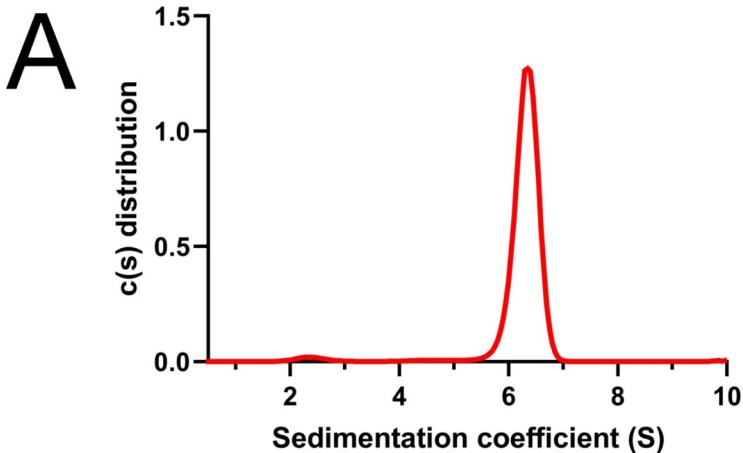

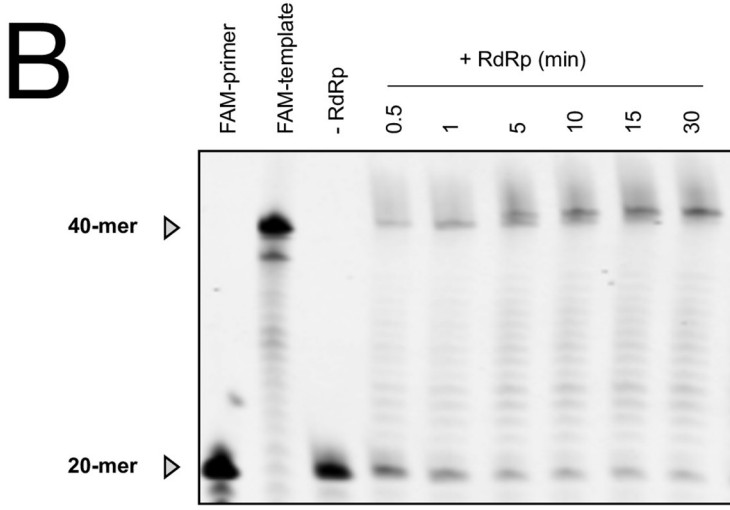

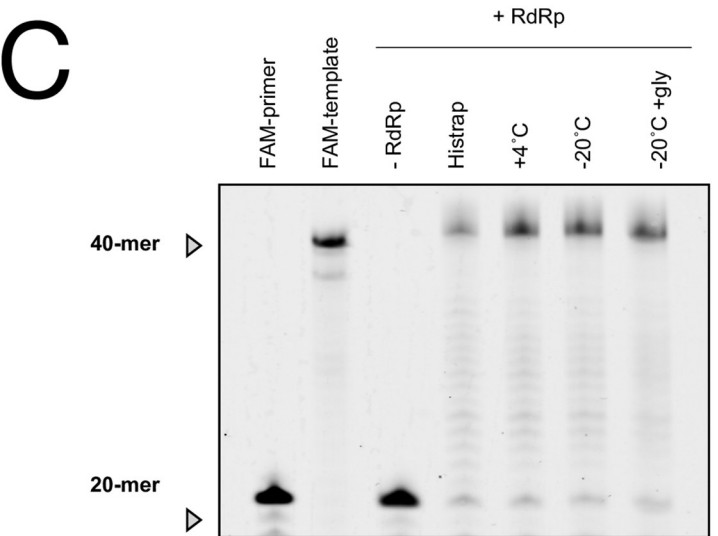

**Fig 4. Biophysical and functional characterization of the recombinant SARS-CoV-2 RdRp complex. (A)** Sedimentation distribution profile of recombinant RdRp complex by sedimentation velocity. The main peak at a concentration of 2.7 mg/mL (16.5 µM) shows a sedimentation coefficients of 6.2 S, compatible with the expected $(nsp7)(nsp8)_2(nsp12)$ stoichiometry. **(B)** RNA primer extension assay with the recombinant RdRp. The recombinant RdRp complex shows polymerase activity in vitro, extending a 20-mer primer strand labeled with a fluorescent dye at the 5′ end (FAM-primer). A 5'-labeled template has been also loaded as a control (FAM-template). **(C)** RNA primer extension assay with various purified RdRp samples. Reactions were carried out for 30 min with RdRp eluted from nickel affinity chromatography (Histrap), with purified RdRp kept at +4°C (+4°C), with purified RdRp flash-frozen in liquid nitrogen and stored at -20°C (-20°C), and with purified RdRp flash-frozen in liquid nitrogen with 50% glycerol and stored at -20°C (-20°C +gly).

bacterial lysate by using one single purification step (nickel affinity), thus facilitating high-throughput screens and biochemical studies. Furthermore, the three-step purification protocol yields an intact and correctly assembled $(nsp7)(nsp8)_2(nsp12)$ complex that should be suitable for many applications, including structural studies. Our construction has been made available to the entire scientific community through the Addgene plasmid repository (Addgene ID: 165451) upon publication of this manuscript.

## Supporting information

**S1 File.**
(PDF)

## Acknowledgments

We would like to thank Dr. Bertrand Raynal (Molecular Biophysics Platform, C2RT, Institut Pasteur) for helping us with AUC data analysis. We wish also to thank Dr. Margarida Gomes for helpful advices for molecular biology.

## Author Contributions

**Conceptualization:** Clément Madru, Ludovic Sauguet, Marc Delarue.

**Funding acquisition:** Ludovic Sauguet, Marc Delarue.

**Investigation:** Clément Madru, Ayten Dizkirici Tekpinar, Sandrine Rosario, Dariusz Czernecki, Sébastien Brûlé.

**Methodology:** Ludovic Sauguet.

**Project administration:** Clément Madru.

**Supervision:** Marc Delarue.

**Validation:** Ludovic Sauguet, Marc Delarue.

**Writing – original draft:** Clément Madru.

**Writing – review & editing:** Clément Madru, Ayten Dizkirici Tekpinar, Sébastien Brûlé, Ludovic Sauguet, Marc Delarue.

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
