## [Decision Letter · Decision Letter 0]

14 Sep 2020

PONE-D-20-24456

Fast and efficient purification of SARS-CoV-2 RNA dependent RNA polymerase complex expressed in Escherichia coli

PLOS ONE

Dear Dr. Sauguet,

Thank you for submitting your manuscript to PLOS ONE. After careful consideration, we feel that it has merit but does not fully meet PLOS ONE’s publication criteria as it currently stands. Therefore, we invite you to submit a revised version of the manuscript that addresses the points raised during the review process.

Please find the reviewers' comments appended below. While all reviewers agree that the work presented in this paper could have a high impact and the potential to significantly facilitate future drug discovery studies on SARS-CoV-2, they have raised points that need to be addressed. Overall, I think very useful and detailed comments have been provided by the reviewers. 

We look forward to receiving your revised manuscript.

Kind regards,

Filippo Prischi

Academic Editor

PLOS ONE

Journal Requirements:

3.

PLOS ONE now requires that authors provide the original uncropped and unadjusted images underlying all blot or gel results reported in a submission’s figures or Supporting Information files. This policy and the journal’s other requirements for blot/gel reporting and figure preparation are described in detail at https://journals.plos.org/plosone/s/figures#loc-blot-and-gel-reporting-requirements and https://journals.plos.org/plosone/s/figures#loc-preparing-figures-from-image-files. When you submit your revised manuscript, please ensure that your figures adhere fully to these guidelines and provide the original underlying images for all blot or gel data reported in your submission. See the following link for instructions on providing the original image data: https://journals.plos.org/plosone/s/figures#loc-original-images-for-blots-and-gels.

Reviewers' comments:

Reviewer's Responses to Questions

**Comments to the Author**

1. Is the manuscript technically sound, and do the data support the conclusions?

Reviewer #1: No

Reviewer #2: Yes

Reviewer #3: No

2. Has the statistical analysis been performed appropriately and rigorously? 

Reviewer #1: N/A

Reviewer #2: Yes

Reviewer #3: N/A

3. Have the authors made all data underlying the findings in their manuscript fully available?

Reviewer #1: Yes

Reviewer #2: Yes

Reviewer #3: Yes

4. Is the manuscript presented in an intelligible fashion and written in standard English?

Reviewer #1: Yes

Reviewer #2: Yes

Reviewer #3: Yes

5. Review Comments to the Author

Reviewer #1: The manuscript by Madru et al. describe a new strategy to purify SARS-CoV-2 RdRp (nsp7/82/12) in E. coli and provide biochemical characterization to demonstrate the quality of the purified, recombinantly expressed RdRp. This study outlines a fast purification scheme for the RdRp which may have applications in general RdRp research and RdRp inhibitor drug discovery. The study uses a series of chromatographic steps to purify the RdRp followed by analytical centrifugation, negative-stain electron microscopy (EM), and primer extension assays to test the quality of the RdRp.

I do feel that a pipeline that would lead to a high yield of the holo-RdRp ((nsp7/82/12) expressed in bacteria would greatly facilitate the efforts many researches trying to screen for inhibitors, so the significance is high and I appreciate's this group's intent. However, the data here do not support that the authors do indeed get a higher yield of protein that the other protocols using bacterial expression data. Indeed, 100 µg from 1.5 L seems rather low and the methods used to purify are not that different from already published methods.

Major critiques

1. The purification described in the paper is largely similar to that described in:

Chen J, Malone B, Llewellyn E, et al. Structural Basis for Helicase-Polymerase Coupling in the SARS-CoV-2 Replication-Transcription Complex [published online ahead of print, 2020 Jul 28]. Cell. 2020;S0092-8674(20)30941-7. doi:10.1016/j.cell.2020.07.033

2. Authors do not reference the appropriate papers and citations are misplaced throughout the manuscript. The following are a few examples out of many where citations are lacking, incorrect, or misplaced:

i. Page 3 - “So far, Remdesivir, Favipiravir, Ribavirin, Galidesivir, and EIDD-2801, have been shown to efficiently inhibit SARS-CoV-2 replication in cell-based assays (14–17) but their efficiency in humans remains to be assessed, rendering the search for new inhibitors still of interest.”

- Reference 17 does not have relevance to this statement.

ii. Page 3 – “RdRp is composed of 3 viral non-structural proteins (nsp) named nsp7, nsp8 and nsp12. The core component nsp12 hosts the catalytic polymerase activity (18) which is greatly enhanced by the two accessory subunits nsp7 and nsp8 (8,19).”

- Please cite Kirchdoerfer and Ward, 2019 and Reference 18 (Subissi et al., 2014) since these studies were the first to demonstrate the importance of nsp7 and nsp8 for nsp12 activity. Remove references 8 and 19 since they are not relevant here.

iii. Page 3 – “Recently, RdRp has been a subject of intensive structural biology efforts, yielding high resolutions cryo-EM structures of the RdRp apo form (20,21), bound with RNA (22,23), and with inhibitors (23,24).”

- Reference 21 is misplaced. The structure presented in this reference contains RNA. Remove reference 24 as it does not pertain to the sentence. Also add “bound to other factors” (cite Chen et al. 2020).

iv. Page 3 – “The production of sufficient amounts of heterologous RdRp with a native structure and full biological activity is a prerequisite for the discovery, optimization and comprehensive evaluation of new drugs directed against SARS-CoV-2, including in High Throughput Screening (HTS) assays involving very large chemical libraries (25).”

- Reference 25 is about the structure PolD complexes and does not have any relevance to HTS.

v. Page 3 – “The main strategies employed so far for the overexpression of recombinant RdRp consists in expressing and purifying the 3 subunits separately before assembling the complex in vitro (19,22).”

- References 19 through 22 all purify nsp7, nsp8, and nsp12 separately to and reconstitute the RdRp in vitro. Please add these citations to this statement. Also Cite Chen et al. 2020 as well as they also purified separately and reconstituted.

vi. Page 3 – “Moreover, while nsp7 and nsp8 express readily in Escherichia coli, nsp12 shows limited solubility in bacterial expression systems and is often produced in insect cells (20,22,24).”

- Reference 24 does not belong here. Cite Chen et al. 2020 as well since they use E. coli to express these proteins.

vii. Page 9 – “Recent structural studies have revealed that this N-terminal region of nsp8 is flexible and gets ordered when the RNA duplex exits from the enzyme’s active site (22).”

- Please cite Reference 21 (Hillen et al. 2020) instead of Reference 22 as Hillen et al. made the first structural observations that the N-terminus of nsp8 contacts the upsteam RNA.

viii. Page 9 – “Yet, this region is not required for RNA polymerase activity but improves its processivity by perpetuating the interactions with the RNA backbone (22).”

- Cite Reference 18 (Subissi et al., 2014) not Reference 22 since Subissi et al. showed that nsp7 and nsp8 are important for polymerase processivity.

ix. Page 14 – “(C) Two orthogonal views of the cryo-EM structure of the SARS-CoV-2 RdRp (PDB code: 6YYT) (22).”

- The reference cited for PDB 6YYT is wrong. Please correct to Reference 21 (Hillen et al. 2020).

3. Authors claim to have a yield of “~100 μg pure RdRp complex” (Page 9) after size exclusion chromatography. However, the sample is not “pure” as it contains a significant proteolyzed product (nsp8-CTD) that is clearly visible after size exclusion chromatography (Figure 2D).

4. Authors claim that “the complex [purified RdRp] has an extended shape, consistent with the RdRp structure” (Page 9) and references Figure 1C and cites References 21 and 22. However, these structures are extended due to the presence of a duplexed RNA scaffold. Analytical ultracentrifugation experiments were performed on the “purified RdRp” (Page 9) and thus lack RNA. Can authors address this discrepancy? In addition, the expected size of an intact RdRp is predicted to be 163 kDa but the authors report 145 kDa (Figure 3A). Why is the purified complex smaller by 18 kDa?

5. Authors perform negative-stain EM to claim that the sample is “homogeneous and not aggregated” (Page 9), however the image presented in Figure 3B is uninformative since the image is low-resolution and the particle population are not shown (no 2D classes). Authors should show that the particles observed by negative-stain EM are in fact intact RdRp particles by reconstructing a 3D volume from the particles. Additionally, the negative stain EM sample was prepared at 0.05 mg/mL (which is magnitudes lower than the concentrations used in cryo-EM and even some biochemical assays) so it is unsurprising that the sample is “not aggregated” (Page 9).

6. The activity of the purified RdRp (Figure 3C) is much weaker compared to reconstituted RdRp (combining nsp12 with nsp7/8, see primer extension assay in Figure 1 of Hillen et al. 2020). Even after 60 mins of incubation, the purified RdRp does not extend all the 20mer primer RNA. Despite this, the authors claim the purified RdRp is “functionally active” (Page 10).

Minor comments

1. Page 2, Abstract – “Characterization of the purified recombinant SARS-Cov-2 shows that it forms a tetramer with the expected stoichiometry.”

- Insert “RdRp” between “SARS-CoV-2” and “shows”.

2. Page 4 – “Here, we describe an alternative strategy to produce the SARS-CoV-2 RdRp directly in E. coli, using a single polycistronic construct.”

- This construct is NOT polycistronic as nsp12 and nsp7/8 are expressed on two separate T7 promoters.

3. Page 4 – “The open reading frames (ORF) of the nsp7, nsp8 and nsp12 genes from SARS-CoV-2virus were synthesized commercially by geneArt (Thermo Fisher)”

- Are these genes made using viral codons or were they codon optimized for expression in E. coli?

4. Page 5. Change “chimio-competent” to “chemocomponent”

5. Page 5. Change “1,4 kPa” to “1.4 kPa”

6. Page 6. Change “histrap fractions” to “HisTrap fractions”

7. Page 6. Change “heparin hiTrap HP” to “HiTrap Heparin HP”

8. Nsp 12 is a cysteine-rich protein (29 cysteines in primary protein sequence), however buffers used for purification and biochemistry do not contain any reducing agents (BME, DTT, TCEP, etc.). Could the authors justify using oxidizing conditions (no reducing agents present) for their purification and biochemistry?

9. The RNA duplex was prepared in water and annealed by only “heating for 2 min at 70 °C” (Page 7) followed by slow cooling to room temperature. Could the authors justify annealing the RNA in water (absence of salt or buffer) and using a lower denaturation temperature?

10. Page 8 – “A polycistronic plasmid was employed to co-express the nsp7, nsp8 and nsp12 subunits of the SARS-CoV-2 RdRp in E. coli.”

- Incorrect usage of the word polycistronic. Nsp12 and nsp7/8 are made from two separate T7 promoters.

Reviewer #2: The Severe Acute Respiratory Syndrome Coronavirus 2 (SARS-CoV-2) causes the current COVID-19 pandemic. The RNA dependent RNA polymerase (RdRP), consisting of nsp7, nsp8, and nsp12, is the key enzyme of the SARS-CoV-2 and a very good antiviral target. Currently, they have been multiple studies about the function and structural studies of the RdRP. The existing strategies used for the overproduction of RdRp involve the expression and purification the 3 subunits separately before assembling the complex in vitro. This paper describes an alternative strategy of using a polycistronic construct to produce the SARS-CoV-2 RdRp in E. coli.

Overall, it is a good paper, but the following questions need to be addressed.

1. Provide the plasmid sequence.

2. Page 12:” The complex was eluted using a 25 mL linear gradient of imidazole (Buffer B: 50 mM Tris-HCl at pH 8, 500 mM NaCl, 500 mM imidazole).” From Fig. 2A, it should be 50ml.

3. The authors should provide the 2D classification of Negative stain images, and the defocus at -3 µm is a little high. And the NS-EM image Fig. 3B shows it has some aggregation.

4. The authors should provide the data of Matrix-Assisted Laser Desorption/Ionization (MALDI-TOF/TOF) analysis.

5. The image of the RNA Primer extension assays (Fig. 3C) should be clearly labeled, especially the control at 0 min and protein only and RNA only. Besides, more assays designed is better.

6. There has another band in the first lane of Fig. 3C, what is that?

7. Pg. 10, “it forms a tetramer with the expected stoichiometry.” This seems vague, perhaps state what the “expected stoichiometry” is.

8. There seems to be a typo on the first line of page 14 (the “and” should be before the MgCl2, not after).

9. They perhaps need to expand on the functions of NSP7 and NSP8 cofactors in the complex instead of just NSP12.

10. This is probably not as important, but they specifically mention that they incubated the cells with agitation (180rpm, pg 11) during the induction step but not in the previous step(s) when they were culturing the cells.

11. Also, a plasmid map would be beneficial in visualizing the final vector.

Reviewer #3: Summary:

This work by Madru et al needs major revision if it is to be published. The authors claim that they have constructed a polycistronic expression system is false as the design utilizes two separate promoters that are regulated independently. The methods used to both express and purify the RdRp complex require further development as the purification has numerous steps that could be optimized.

The authors incorrectly reference other papers throughout the manuscript which indicates a lack of attention to detail or knowledge about the field.

Plos one paper review:

• This is not actually polycistronic – its a duet expression vector with two separate T7 promoters. See methods.

There is evidence in support of Remdesivir's use in humans (it is incorrect to say that their efficiency in humans remains to be tested) as there have been several clinical trials conducted.

References starting at reference 19 are off by one reference.

“Recently, RdRp has been a subject of intensive structural biology efforts, yielding high resolutions cryo-EM structures of the RdRp apo form(20,21), bound with RNA (22,23), and with inhibitors (23,24)”

Gao et al is numbered reference 19 in the references section. Please change this line to:

“Recently, RdRp has been a subject of intensive structural biology efforts, yielding high resolutions cryo-EM structures of the RdRp apo form(19,20), bound with RNA (21,22), and with inhibitors (22,23)”

In ref 19, the authors co-expressed nsp7 and nsp8 which has been done in other recent manuscripts Chen, Malone et al, 2020. The statement “The main strategies employed so far for the overexpression of recombinant RdRp consists in expressing and purifying the 3 subunits separately before assembling the complex in vitro” is incorrect.

“Moreover, while nsp7 and nsp8 express readily in Escherichia coli, nsp12 shows limited solubility in bacterial expression systems and is often produced in insect cells (20,22,24).” Wang et al use insect cell expression, this statement is likely made to reference Peng and Hillen et al.

“These approaches lengthen the protein expression and purification steps” would make better sense than “These approaches multiply the protein expression and purification steps”

Methods:

Authors need to state if ORFs were codon optimized, and how.

The design is not polycistronic. Polycistronic implies that the three genes are transcribed as a single transcript that has separate RBS sites (there are two T7 promoters so two mRNA transcripts in this design).

Given use of codon plus BL21 cells, I am assuming that the ORFs of nsp12/7/8 are not codon optimized. Please confirm as mentioned in previous point if these cells are necessary.

Protein expression:

Please elaborate what is LBKC. Was this supposed to say ‘LB’?

Is 1 mM IPTG correct for overnight expression?

‘conserved at -80’ – Use of ‘kept’ or ‘stored’ would be better.

Purification:

Would doing the heparin step pre the nickel purification step be better? The authors state that there was nucleic acid contamination of the Nickel elution. Do the authors lose protein in the flow-through for the nickel step due to poorer binding of the nucleic acid bound RdRp complex to the Ni-NTA?

Please rephrase description of the protein purification. The use of Nickel buffer A, Nickel Buffer B, Heparin buffer A, Heparin buffer B, SD200 buffer as descriptors for the purification buffers maybe more reader friendly. The authors should also specify reducing agents that were utilized in each of these buffers.

Can the authors comment on the need to use the Q column post the use of a heparin column? Have the authors compared running the heparin column over a 20CV gradient and selecting only those fractions containing nsp12?

“The purification was finally polished using..” - Please re-phrase.

The authors should specify if they add glycerol prior to flash-freezing.

RNA primer extension assays:

Statement reads:

“The primer extension assay was performed with 500 nM of purified RdRp complex in the presence of 500 μM NTPs and 100 nM RNA duplex, in a reaction buffer containing 20 mM Na-Hepes pH 8, 50 mM NaCl, 3 mM MgCl2 and.”

Please remove the ‘and.’ or complete the sentence.

Results and discussion:

“The UV spectra of the eluted fractions showed a large DNA contamination with A260/A280 ratio of 1.4 from this stage” – Insert ‘an’ after ‘with - Replace ‘from’ with ‘at’

“Finally, the purification was polished using a size-exclusion chromatography showing one single peak” – Remove ‘polished’ with ‘completed” and ‘showing’ with ‘which showed’

“Recent structural studies have revealed that this N terminal region of nsp8 is flexible and gets ordered when the RNA duplex exits from the enzyme’s active site (22).” – Rephrase “gets ordered”. Please cite Hillen et al for this statement.

“Yet, this region is not required for RNA polymerase activity but improves its processivity by perpetuating the interactions with the RNA backbone (22)” -Please rephrase. This region is required for polymerase activity as determined by the phenotypic reduced viral replication in the presence of nsp8 K58 mutations as shown in subissi et al, 2014. The use of ‘improves’ implies a non-lethal phenotype which is not the case.

“An overall yield of ~100 µg pure RdRp complex was obtained from 1 g of E. coli pellet after the final size exclusion chromatography.” – It is custom to give the yield as mg / L of culture if specifying at all.

Biophysical and functional characterization…

“In addition, the purified RdRp complex was applied onto glowdischarged carbon coated EM grids, and stained with a 2% uranyl formate aqueous solution. Images were recorded with a defocus of -3�m, at the instrument magnification of 49000” –The underlined above is redundant since stated in the methods.

Figures:

Fig 1

The reader would require a higher resolution image first and foremost.

The authors use fig 1 B (i.e pdb 6YYT from Hillen et al which is cited incorrectly as wang et al ) as a discussion piece for why their AUC frictional coefficient was 1.5. The authors are purifying the RdRp complex in the absence of RNA. They have correctly stated that the N-terminus of nsp8 is ordered in the presence of upstream RNA. In its absence, the N terminus is disordered and the resultant comment that the authors make regarding an “extended shape” in relation to the AUC data needs to be further developed.

The structure is also quite pixelated.

Fig 2

The reader would require a higher resolution image first and foremost.

In relation to the Nickel purification step, the authors should comment on the need to include 500 mM imidazole in nickel buffer B. It is more common when using a nickel column to do step wise washes of 20-50 mM imidazole prior to elution with 250 mM imidazole.

Fig 3

The reader would require a higher resolution image first and foremost.

Not sure what information is being provided by the negative stain image. The AUC shows the absence of aggregation on its own.

6. PLOS authors have the option to publish the peer review history of their article (what does this mean?). If published, this will include your full peer review and any attached files.

Reviewer #1: No

Reviewer #2: No

Reviewer #3: No

---

## [Author Response · Author response to Decision Letter 0]

11 Mar 2021

REVIEWER #1:

The manuscript by Madru et al. describe a new strategy to purify SARS-CoV-2 RdRp (nsp7/82/12) in E. coli and provide biochemical characterization to demonstrate the quality of the purified, recombinantly expressed RdRp. This study outlines a fast purification scheme for the RdRp which may have applications in general RdRp research and RdRp inhibitor drug discovery. The study uses a series of chromatographic steps to purify the RdRp followed by analytical centrifugation, negative-stain electron microscopy (EM), and primer extension assays to test the quality of the RdRp.

I do feel that a pipeline that would lead to a high yield of the holo-RdRp ((nsp7/82/12) expressed in bacteria would greatly facilitate the efforts many researches trying to screen for inhibitors, so the significance is high and I appreciate's this group's intent. However, the data here do not support that the authors do indeed get a higher yield of protein that the other protocols using bacterial expression data. Indeed, 100 µg from 1.5 L seems rather low and the methods used to purify are not that different from already published methods.

 We understand the reviewer’s concerns. Therefore, while preparing this revised version of the manuscript, we have designed a new plasmid named pRSFDuet-1(14his-nsp8/nsp7)(nsp12) and a new strategy to purifiy RdRP. With this new strategy, we are able to produce a full-length complex by fusing the 14His tag to the N-terminus of nsp8. With this new construct, the purification yield has been increased passing 5-fold 0.2 to 1 mg of complex per liter of culture, and no proteolysis is observed. The new purification protocol has been improved and simplified. Consistently, the biophysical and functional characterization of this newly purified complex has been improved. In the revised manuscript, we have shown that active RdRP can be obtained from one single affinity chromatography step, giving to this plasmid far-reaching potential in high throughput screening methods. 

We sincerely apologize for our mistakes in the numbering of the references. We wanted this new construct to be available as quickly as possible, to help the scientific community that is fighting against covid-19 and did not carefully enough check the accuracy of the references. 

 Major critiques

1. The purification described in the paper is largely similar to that described in:

Chen J, Malone B, Llewellyn E, et al. Structural Basis for Helicase-Polymerase Coupling in the SARS-CoV-2 Replication-Transcription Complex [published online ahead of print, 2020 Jul 28]. Cell. 2020;S0092-8674(20)30941-7. doi:10.1016/j.cell.2020.07.033

Even though the purification protocol is similar, our method allows the RdRp production from one single plasmid. The table below describes the recombinant expression system used to date for RdRp production in the literature. To our knowledge, our approach is unique and never published so far. This offers specific advantages that are now better described in the current version of the manuscript. With our novel construct, we are also able to obtain pure and active RdRP following one single Ni-NTA purification step.

 NSP12 NSP8 NSP7

Yin et al. https://science.sciencemag.org/content/368/6498/1499

inssect cells

 E.coli

Wang et al. https://doi.org/10.1016/j.cell.2020.05.034

E.coli E.coli E.coli

Hillen et al.

Kokic et al. https://www.nature.com/articles/s41586-020-2368-8

https://www.ncbi.nlm.nih.gov/pmc/articles/PMC7804290/ inssect cells E.coli E.coli

Gao et al.

Yan et al.

Yan et al. https://science.sciencemag.org/content/368/6492/779

https://www.ncbi.nlm.nih.gov/pmc/articles/PMC7675986/

https://www.sciencedirect.com/science/article/pii/S0092867420315336?via%3Dihub#fig3 E.coli E.coli

Peng et al. https://doi.org/10.1016/j.celrep.2020.107774

inssect cells E.coli E.coli

 nsp7L8 fusion in E.coli

Gordon et al.

Tchesnokov et al. https://www.jbc.org/content/295/20/6785

https://www.jbc.org/content/early/2020/09/23/jbc.AC120.015720

inssect cells (polyprotein with nsp5)

Shannon et al. https://www.ncbi.nlm.nih.gov/pmc/articles/PMC7499305/

E.coli E.coli E.coli

 nsp7L8 fusion in E.coli

Chen et al. 10.1016/j.cell.2020.07.033

E.coli E.coli

Naydenova et al. https://www.ncbi.nlm.nih.gov/pmc/articles/PMC7896311/ inssect cells (polyprotein with nsp5)

 E.coli

Chien et al. 

Jockusch et al. 10.1021/acs.jproteome.0c00392

10.1016/j.antiviral.2020.104857

inssect cells E.coli E.coli

Li et al. https://www.ncbi.nlm.nih.gov/pmc/articles/PMC7883726/ E.coli E.coli

Dangerfield et al. https://www.sciencedirect.com/science/article/pii/S2589004220310464

https://www.ncbi.nlm.nih.gov/pmc/articles/PMC7859716/ E. coli + chaperones

 E. coli 

2. Authors do not reference the appropriate papers and citations are misplaced throughout the manuscript. The following are a few examples out of many where citations are lacking, incorrect, or misplaced:

i. Page 3 - “So far, Remdesivir, Favipiravir, Ribavirin, Galidesivir, and EIDD-2801, have been shown to efficiently inhibit SARS-CoV-2 replication in cell-based assays (14–17) but their efficiency in humans remains to be assessed, rendering the search for new inhibitors still of interest.”

- Reference 17 does not have relevance to this statement.

This sentence has been removed.

ii. Page 3 – “RdRp is composed of 3 viral non-structural proteins (nsp) named nsp7, nsp8 and nsp12. The core component nsp12 hosts the catalytic polymerase activity (18) which is greatly enhanced by the two accessory subunits nsp7 and nsp8 (8,19).”Please cite Kirchdoerfer and Ward, 2019 and Reference 18 (Subissi et al., 2014) since these studies were the first to demonstrate the importance of nsp7 and nsp8 for nsp12 activity. Remove references 8 and 19 since they are not relevant here.

Corrected and added.

iii. Page 3 – “Recently, RdRp has been a subject of intensive structural biology efforts, yielding high resolutions cryo-EM structures of the RdRp apo form (20,21), bound with RNA (22,23), and with inhibitors (23,24).”

- Reference 21 is misplaced. The structure presented in this reference contains RNA. Remove reference 24 as it does not pertain to the sentence. Also add “bound to other factors” (cite Chen et al. 2020).

Corrected and added.

iv. Page 3 – “The production of sufficient amounts of heterologous RdRp with a native structure and full biological activity is a prerequisite for the discovery, optimization and comprehensive evaluation of new drugs directed against SARS-CoV-2, including in High Throughput Screening (HTS) assays involving very large chemical libraries (25).”Reference 25 is about the structure PolD complexes and does not have any relevance to HTS.

Removed.

v. Page 3 – “The main strategies employed so far for the overexpression of recombinant RdRp consists in expressing and purifying the 3 subunits separately before assembling the complex in vitro (19,22).”

- References 19 through 22 all purify nsp7, nsp8, and nsp12 separately to and reconstitute the RdRp in vitro. 

Please add these citations to this statement. Also Cite Chen et al. 2020 as well as they also purified separately and reconstituted.

Corrected and added.

vi. Page 3 – “Moreover, while nsp7 and nsp8 express readily in Escherichia coli, nsp12 shows limited solubility in bacterial expression systems and is often produced in insect cells (20,22,24).”

- Reference 24 does not belong here. Cite Chen et al. 2020 as well since they use E. coli to express these proteins.

Corrected and added.

vii. Page 9 – “Recent structural studies have revealed that this N-terminal region of nsp8 is flexible and gets ordered when the RNA duplex exits from the enzyme’s active site (22).”

- Please cite Reference 21 (Hillen et al. 2020) instead of Reference 22 as Hillen et al. made the first structural observations that the N-terminus of nsp8 contacts the upsteam RNA.

Corrected.

viii. Page 9 – “Yet, this region is not required for RNA polymerase activity but improves its processivity by perpetuating the interactions with the RNA backbone (22).”

- Cite Reference 18 (Subissi et al., 2014) not Reference 22 since Subissi et al. showed that nsp7 and nsp8 are important for polymerase processivity.

This sentence has been removed.

ix. Page 14 – “(C) Two orthogonal views of the cryo-EM structure of the SARS-CoV-2 RdRp (PDB code: 6YYT) (22).”

- The reference cited for PDB 6YYT is wrong. Please correct to Reference 21 (Hillen et al. 2020).

This figure has been removed.

3. Authors claim to have a yield of “~100 μg pure RdRp complex” (Page 9) after size exclusion chromatography. However, the sample is not “pure” as it contains a significant proteolyzed product (nsp8-CTD) that is clearly visible after size exclusion chromatography (Figure 2D).

This is not true anymore with our new construct that is no longer proteolyzed and from which we can obtain higher yields. The new plasmid construct yields full-length native complex.

4. Authors claim that “the complex [purified RdRp] has an extended shape, consistent with the RdRp structure” (Page 9) and references Figure 1C and cites References 21 and 22. However, these structures are extended due to the presence of a duplexed RNA scaffold. Analytical ultracentrifugation experiments were performed on the “purified RdRp” (Page 9) and thus lack RNA. Can authors address this discrepancy? In addition, the expected size of an intact RdRp is predicted to be 163 kDa but the authors report 145 kDa (Figure 3A). Why is the purified complex smaller by 18 kDa?

First of all, we aimed here to demonstrate the homogeneity of the purified complex and not to accurately measure the molecular weight (MW). However, the apparent MW of 145kDa was indeed lower than the theoretical value (111 + 22 + 22 + 9 = 164 kDa with 14His tag). This discrepancy could be explained by two main reasons: (i) the pRSFDuet-1(14his-nsp12)(nsp7/nsp8) yielded a partially proteolyzed complex, lacking the ~10 kDa nsp8 N-terminal region, (ii) the weak contribution of the ~4.5 kDa 14His TEV-cleavable tag to the sedimentation. 

Our new construct, pRSFDuet-1(14his-nsp8/nsp7)(nsp12), described in the current manuscript version allows the purification of the whole complex, without proteolysis. The new AUC sedimentation profile exhibits one main peak, corresponding to a 160 kDa complex much closer to the theoretical value (106.5 + 26.5 + 26.5 + 9 = 168.5 with two 14-His tag). We attribute this ~5% error to the low contribution of the two 14His flexible tags to the sedimentation. 

In addition, the sentence “the complex [purified RdRp] has an extended shape, consistent with the RdRp structure” has been replaced by “the frictional ratio of 1.55 suggests that the heterotetramer is slightly elongated”. 

5. Authors perform negative-stain EM to claim that the sample is “homogeneous and not aggregated” (Page 9), however the image presented in Figure 3B is uninformative since the image is low-resolution and the particle population are not shown (no 2D classes). Authors should show that the particles observed by negative-stain EM are in fact intact RdRp particles by reconstructing a 3D volume from the particles. Additionally, the negative stain EM sample was prepared at 0.05 mg/mL (which is magnitudes lower than the concentrations used in cryo-EM and even some biochemical assays) so it is unsurprising that the sample is “not aggregated” (Page 9).

Following the reviewer’s #1 comment, and as suggested by the reviewer #3, we have decided to remove the negative-stain EM image from the manuscript. Indeed, the present study aims to propose a production method and not a structural study. The current protocol yields a pure complex, biochemically and biophysically characterized through a 3-step quality control showing (i) full-length subunits on SDS-PAGE analysis, (ii) absence of aggregates in AUC assay (iii) polymerase activity on primer extension assays. In our opinion, reconstructing an EM 3D model using our purified RdRP is beyond the scope of this paper. 

6. The activity of the purified RdRp (Figure 3C) is much weaker compared to reconstituted RdRp (combining nsp12 with nsp7/8, see primer extension assay in Figure 1 of Hillen et al. 2020). Even after 60 mins of incubation, the purified RdRp does not extend all the 20mer primer RNA. Despite this, the authors claim the purified RdRp is “functionally active” (Page 10).

The functional assay has been repeated more than 3 times with the newly purified RdRp, and have shown better results (see new Fig 4B and 4C). The full-length product is now detected after only 30 seconds at 37˚C, with a strong signal that stabilizes after 5 min. 

Minor comments

1. Page 2, Abstract – “Characterization of the purified recombinant SARS-Cov-2 shows that it forms a tetramer with the expected stoichiometry.”

- Insert “RdRp” between “SARS-CoV-2” and “shows”.

Corrected.

2. Page 4 – “Here, we describe an alternative strategy to produce the SARS-CoV-2 RdRp directly in E. coli, using a single polycistronic construct.”

- This construct is NOT polycistronic as nsp12 and nsp7/8 are expressed on two separate T7 promoters.

We agree with the reviewer #1. The word polycistronic has been removed from the manuscript.

3. Page 4 – “The open reading frames (ORF) of the nsp7, nsp8 and nsp12 genes from SARS-CoV-2virus were synthesized commercially by geneArt (Thermo Fisher)”

- Are these genes made using viral codons or were they codon optimized for expression in E. coli?

The ORFs of nsp7, nsp8 and nsp12 were optimized for expression in E. coli. it is now specified in the materiel and method section. 

4. Page 5. Change “chimio-competent” to “chemocomponent”

Done.

5. Page 5. Change “1,4 kPa” to “1.4 kPa”

Corrected.

6. Page 6. Change “histrap fractions” to “HisTrap fractions”

Corrected.

7. Page 6. Change “heparin hiTrap HP” to “HiTrap Heparin HP”

Corrected.

8. Nsp12 is a cysteine-rich protein (29 cysteines in primary protein sequence), however buffers used for purification and biochemistry do not contain any reducing agents (BME, DTT, TCEP, etc.). Could the authors justify using oxidizing conditions (no reducing agents present) for their purification and biochemistry? `

Considering reviewer #1’s comment, we have repeated the purification in presence of 5 mM DTT in all buffers. Results were identical. 

9. The RNA duplex was prepared in water and annealed by only “heating for 2 min at 70 °C” (Page 7) followed by slow cooling to room temperature. Could the authors justify annealing the RNA in water (absence of salt or buffer) and using a lower denaturation temperature?

We have calculated the melting temperature of the RNA primer which is 58.5˚C. Heating for 2 min at 70˚C should be therefore good enough to denature and re-hybridize the RNA duplex. In addition, we perform RNA duplex denaturation and renaturation in water to avoid hydrolysis promoted by magnesium ions (See AbouHaidar MG, Ivanov IG. Non-Enzymatic RNA Hydrolysis Promoted by the Combined Catalytic Activity of Buffers and Magnesium Ions. Zeitschrift für Naturforschung C. 1999 Aug 1;54(7–8):542–8.). 

10. Page 8 – “A polycistronic plasmid was employed to co-express the nsp7, nsp8 and nsp12 subunits of the SARS-CoV-2 RdRp in E. coli.”

- Incorrect usage of the word polycistronic. Nsp12 and nsp7/8 are made from two separate T7 promoters.

We agree with the reviewer #1. The word polycistronic has been removed from the manuscript.

REVIEWER #2:

The Severe Acute Respiratory Syndrome Coronavirus 2 (SARS-CoV-2) causes the current COVID-19 pandemic. The RNA dependent RNA polymerase (RdRP), consisting of nsp7, nsp8, and nsp12, is the key enzyme of the SARS-CoV-2 and a very good antiviral target. Currently, they have been multiple studies about the function and structural studies of the RdRP. The existing strategies used for the overproduction of RdRp involve the expression and purification the 3 subunits separately before assembling the complex in vitro. This paper describes an alternative strategy of using a polycistronic construct to produce the SARS-CoV-2 RdRp in E. coli.

Overall, it is a good paper, but the following questions need to be addressed.

1. Provide the plasmid sequence.

We have added the full annotated plasmid sequence in the supplementary Figure 2. It is also available online on the Addgene website.

2. Page 12:” The complex was eluted using a 25 mL linear gradient of imidazole (Buffer B: 50 mM Tris-HCl at pH 8, 500 mM NaCl, 500 mM imidazole).” From Fig. 2A, it should be 50ml.

Corrected.

3. The authors should provide the 2D classification of Negative stain images, and the defocus at -3 µm is a little high. And the NS-EM image Fig. 3B shows it has some aggregation.

As explained to the reviewer #1 (5th point), we have decided to remove the negative staining EM Figure in the revised manuscript.

4. The authors should provide the data of Matrix-Assisted Laser Desorption/Ionization (MALDI-TOF/TOF) analysis.

 We agree with the reviewer #2. The identification by mass spectrometry of the additional band should have been shown in the first version of the manuscript. However, this problem is obsolete now with our new construct. Indeed, this fourth band is no longer present with the new protocol and the new construct, we think there is no need to put it in the current version. 

5. The image of the RNA Primer extension assays (Fig. 3C) should be clearly labeled, especially the control at 0 min and protein only and RNA only. Besides, more assays designed is better.

Functional assays have been re-designed, optimized, and clearly labeled (See the new Fig 4).

6. There has another band in the first lane of Fig. 3C, what is that?

This Figure has been replaced. 

7. Pg. 10, “it forms a tetramer with the expected stoichiometry.” This seems vague, perhaps state what the “expected stoichiometry” is.

This sentence has been corrected as follows: “Characterization of the purified recombinant SARS-CoV-2 RdRp shows that it forms a complex with the expected (nsp7)(nsp8)2(nsp12) stoichiometry”.

8. There seems to be a typo on the first line of page 14 (the “and” should be before the MgCl2, not after).

Corrected.

9. They perhaps need to expand on the functions of NSP7 and NSP8 cofactors in the complex instead of just NSP12.

The introduction has been modified and the functions of nsp7 and nsp8 are now better described: “These structures showed the nsp12 core bound to a heterodimer nsp7-nsp8 and an additional nsp8 at a different binding site. The two nsp8 copies expose long N-terminal alpha helices that slide along the exiting RNA to prevent premature dissociation.”. 

10. This is probably not as important, but they specifically mention that they incubated the cells with agitation (180rpm, pg 11) during the induction step but not in the previous step(s) when they were culturing the cells.

Agitation conditions are now specified along the protocol. 

11. Also, a plasmid map would be beneficial in visualizing the final vector.

We have added the full annotated plasmid map in the supplementary Figure 1.

REVIEWER #3:

This work by Madru et al needs major revision if it is to be published. The authors claim that they have constructed a polycistronic expression system is false as the design utilizes two separate promoters that are regulated independently. The methods used to both express and purify the RdRp complex require further development as the purification has numerous steps that could be optimized. The authors incorrectly reference other papers throughout the manuscript which indicates a lack of attention to detail or knowledge about the field.

We understand the reviewer’s concerns. Therefore, while preparing this revised version of the manuscript, we have designed a new plasmid named pRSFDuet-1(14his-nsp8/nsp7)(nsp12) and a new strategy to purifiy RdRP. With this new strategy, we are able to produce a full-length complex by fusing the 14His tag to the N-terminus of nsp8. With this new construct, the purification yield has been increased 5-fold from 0.2 to 1 mg of complex per liter of culture, and no proteolysis is observed. The new purification protocol has been improved and simplified. Consistently, the biophysical and functional characterization of this newly purified complex has been improved. In the revised manuscript, we have shown that active RdRP can be obtained from one single affinity chromatography step, giving to this plasmid far-reaching potential in high throughput screening methods. We sincerely apologize for our mistakes in the numbering of the references. We wanted this new construct to be available as quickly as possible to help the scientific community that is fighting against covid-19 and did not carefully enough check the accuracy of the references. 

1) This is not actually polycistronic – its a duet expression vector with two separate T7 promoters. See methods.

We agree with the reviewer #3. The word polycistronic has been removed from the manuscript. 

2) There is evidence in support of Remdesivir's use in humans (it is incorrect to say that their efficiency in humans remains to be tested) as there have been several clinical trials conducted.

Corrected in the new introduction.

3) References starting at reference 19 are off by one reference.

Corrected.

4) “Recently, RdRp has been a subject of intensive structural biology efforts, yielding high resolutions cryo-EM structures of the RdRp apo form (20,21), bound with RNA (22,23), and with inhibitors (23,24)”

Gao et al is numbered reference 19 in the references section. Please change this line to:

“Recently, RdRp has been a subject of intensive structural biology efforts, yielding high resolutions cryo-EM structures of the RdRp apo form (19,20), bound with RNA (21,22), and with inhibitors (22,23)”

Corrected.

5) In ref 19, the authors co-expressed nsp7 and nsp8 which has been done in other recent manuscripts Chen, Malone et al, 2020. The statement “The main strategies employed so far for the overexpression of recombinant RdRp consists in expressing and purifying the 3 subunits separately before assembling the complex in vitro” is incorrect.

 This sentence has been modified as follows: “The main strategies employed so far for the overexpression of recombinant RdRp consists in expressing and purifying the catalytic nsp12 subunit and the accessory nsp7-nsp8 subunits separately before assembling the complex in vitro”.

6) “Moreover, while nsp7 and nsp8 express readily in Escherichia coli, nsp12 shows limited solubility in bacterial expression systems and is often produced in insect cells (20,22,24).” Wang et al use insect cell expression, this statement is likely made to reference Peng and Hillen et al.

Corrected.

7) “These approaches lengthen the protein expression and purification steps” would make better sense than “These approaches multiply the protein expression and purification steps”

Corrected.

Methods:

8) Authors need to state if ORFs were codon optimized, and how.

Given use of codon plus BL21 cells, I am assuming that the ORFs of nsp12/7/8 are not codon optimized. Please confirm as mentioned in previous point if these cells are necessary.

The ORFs of nsp7, nsp8 and nsp12 were optimized for expression in E. coli. it is now specified in the materiel and method section. 

9) The design is not polycistronic. Polycistronic implies that the three genes are transcribed as a single transcript that has separate RBS sites (there are two T7 promoters so two mRNA transcripts in this design).

We agree with the reviewer #3. The word polycistronic has been removed from the manuscript. 

Protein expression:

10) Please elaborate what is LBKC. Was this supposed to say ‘LB’? 

This sentence has been corrected as follows “was grown overnight in Lysogeny Broth medium

supplemented with 100 μg/mL kanamycin (LBK).”

11) Is 1 mM IPTG correct for overnight expression?

The induction parameters have been optimized (see new Fig 2), including inducer concentration, post-induction time and post-induction temperature.

12) ‘conserved at -80’ – Use of ‘kept’ or ‘stored’ would be better.

Corrected.

Purification:

13) Would doing the heparin step pre the nickel purification step be better? The authors state that there was nucleic acid contamination of the Nickel elution. Do the authors lose protein in the flow-through for the nickel step due to poorer binding of the nucleic acid bound RdRp complex to the Ni-NTA?

There is no longer DNA contamination using the pRSFDuet-1(14his-nsp8/nsp7)(nsp12).

14) Please rephrase description of the protein purification. The use of Nickel buffer A, Nickel Buffer B, Heparin buffer A, Heparin buffer B, SD200 buffer as descriptors for the purification buffers maybe more reader friendly. The authors should also specify reducing agents that were utilized in each of these buffers.

Corrected. 

15) Can the authors comment on the need to use the Q column post the use of a heparin column? Have the authors compared running the heparin column over a 20CV gradient and selecting only those fractions containing nsp12?

The new protocol includes only one step of anion exchange chromatography. 

16) “The purification was finally polished using..” - Please re-phrase.

Corrected: “The purification final step involves..”

17) The authors should specify if they add glycerol prior to flash-freezing.

The new Figure 4C shows the influence of freezing condition on RdRp activity.

RNA primer extension assays:

18) “The primer extension assay was performed with 500 nM of purified RdRp complex in the presence of 500 μM NTPs and 100 nM RNA duplex, in a reaction buffer containing 20 mM Na-Hepes pH 8, 50 mM NaCl, 3 mM MgCl2 and.”Please remove the ‘and.’ or complete the sentence.

Corrected.

Results and discussion:

19) “The UV spectra of the eluted fractions showed a large DNA contamination with A260/A280 ratio of 1.4 from this stage” – Insert ‘an’ after ‘with - Replace ‘from’ with ‘at’

This sentence has been removed.

19) “Finally, the purification was polished using a size-exclusion chromatography showing one single peak” – Remove ‘polished’ with ‘completed” and ‘showing’ with ‘which showed’

Corrected.

20) “Recent structural studies have revealed that this N terminal region of nsp8 is flexible and gets ordered when the RNA duplex exits from the enzyme’s active site (22).” – Rephrase “gets ordered”. Please cite Hillen et al for this statement.

This sentence has been corrected as follows: “This flexible region becomes ordered when the RNA duplex exits from the enzyme’s active site.”

21) “Yet, this region is not required for RNA polymerase activity but improves its processivity by perpetuating the interactions with the RNA backbone (22)” -Please rephrase. This region is required for polymerase activity as determined by the phenotypic reduced viral replication in the presence of nsp8 K58 mutations as shown in subissi et al, 2014. The use of ‘improves’ implies a non-lethal phenotype which is not the case.

This sentence has been removed from the manuscript. 

22) “An overall yield of ~100 µg pure RdRp complex was obtained from 1 g of E. coli pellet after the final size exclusion chromatography.” – It is custom to give the yield as mg / L of culture if specifying at all.

The yield is now given in mg of protein / L of culture

23) Biophysical and functional characterization…

24) “In addition, the purified RdRp complex was applied onto glowdischarged carbon coated EM grids, and stained with a 2% uranyl formate aqueous solution. Images were recorded with a defocus of -3�m, at the instrument magnification of 49000” –The underlined above is redundant since stated in the methods.

These data have been removed. 

Figures:

25) Fig 1

The reader would require a higher resolution image first and foremost.

The authors use fig 1 B (i.e pdb 6YYT from Hillen et al which is cited incorrectly as wang et al ) as a discussion piece for why their AUC frictional coefficient was 1.5. The authors are purifying the RdRp complex in the absence of RNA. They have correctly stated that the N-terminus of nsp8 is ordered in the presence of upstream RNA. In its absence, the N terminus is disordered and the resultant comment that the authors make regarding an “extended shape” in relation to the AUC data needs to be further developed.

The structure is also quite pixelated.

As explained to the reviewer #1 (4th point), the AUC analysis has been improved. 

26) Fig 2

The reader would require a higher resolution image first and foremost.

In relation to the Nickel purification step, the authors should comment on the need to include 500 mM imidazole in nickel buffer B. It is more common when using a nickel column to do step wise washes of 20-50 mM imidazole prior to elution with 250 mM imidazole.

We agree with the reviewer #3. However, as the RdRp produced from the new pRSFDuet-1(14his-nsp8/nsp7)(nsp12) contains two different 14His tags, it is much strongly bound to the Nickel resin. We have thus decided to keep the same imidazole range for the elution step. 

We followed the advice of Reviewer #3 by including a 5% Histrap Buffer B wash step which greatly enhance the chromatogram profile. 

27) Fig 3

The reader would require a higher resolution image first and foremost.

Not sure what information is being provided by the negative stain image. The AUC shows the absence of aggregation on its own.

As explained to the reviewer #1 (5th point), we have decided to remove the negative stain EM from the manuscript.

---

## [Decision Letter · Decision Letter 1]

25 Mar 2021

PONE-D-20-24456R1

Fast and efficient purification of SARS-CoV-2 RNA dependent RNA polymerase complex expressed in Escherichia coli

PLOS ONE

Dear Dr. Sauguet,

Thank you for submitting your manuscript to PLOS ONE. After careful consideration, we feel that it has merit but does not fully meet PLOS ONE’s publication criteria as it currently stands. Therefore, we invite you to submit a revised version of the manuscript that addresses the points raised during the review process.

Reviewers agree that the paper has improved and all comments raised have been properly answered. One reviewer has suggested some very minor editorial changes (see attached document). If you could respond to these 5 comments, I'll then be happy to accept the paper for publication.

We look forward to receiving your revised manuscript.

Kind regards,

Filippo Prischi

Academic Editor

PLOS ONE

Journal Requirements:

Reviewers' comments:

Reviewer's Responses to Questions

**Comments to the Author**

1. If the authors have adequately addressed your comments raised in a previous round of review and you feel that this manuscript is now acceptable for publication, you may indicate that here to bypass the “Comments to the Author” section, enter your conflict of interest statement in the “Confidential to Editor” section, and submit your "Accept" recommendation.

Reviewer #1: All comments have been addressed

Reviewer #2: All comments have been addressed

2. Is the manuscript technically sound, and do the data support the conclusions?

Reviewer #1: (No Response)

Reviewer #2: Yes

3. Has the statistical analysis been performed appropriately and rigorously? 

Reviewer #1: Yes

Reviewer #2: Yes

4. Have the authors made all data underlying the findings in their manuscript fully available?

Reviewer #1: No

Reviewer #2: Yes

5. Is the manuscript presented in an intelligible fashion and written in standard English?

Reviewer #1: Yes

Reviewer #2: Yes

6. Review Comments to the Author

Reviewer #1: please see attached. I submitted as attachment to color code responses and my comments. It's easier as I had to copy and paste text from manuscript as line numbers were not included in the manuscript.

Reviewer #2: I want to thank the authors for addressing my initial comments. Following the revision to the article, I do not have more questions now.

7. PLOS authors have the option to publish the peer review history of their article (what does this mean?). If published, this will include your full peer review and any attached files.

Reviewer #1: No

Reviewer #2: No

---

## [Author Response · Author response to Decision Letter 1]

6 Apr 2021

The revised manuscript by Madru et al. is greatly improved. The significance of obtaining active RdRp rapidly would facilitate high-throughput screens, and the relative ease of obtaining active protein with one step is commendable. I have a few comments that need addressing, but I do recommend acceptance with some revision. Because the manuscript does not have page or line numbers, I have copied and pasted where my comments are addressing: 

1.”The main strategies employed so far for the overexpression of recombinant RdRp

consists in expressing and purifying the catalytic nsp12 subunit and the accessory nsp7-nsp8 subunits separately before assembling the complex in vitro (11,22,24,28–30).”

-The authors, in their response, cited Dangerfield's 2021 work which also employs a co- expression system, so this claim is false. This paper should be cited within this manuscript. This submitted manuscript's advantage is that the preps can be performed in one day and yield active enzymes. I appreciate the authors did initiate these studies before Dangerfield's paper was published. Therefore, they could state while their manuscript was in review, another group published a co-expression system but one that uses at least 4 for five steps. 

This paper is now cited in our manuscript : “While this manuscript was under review, a method for producing tag-free SARS-CoV-2 RdRp in E. coli has been reported by Dangefield et al. (32).Their approach requires many more steps than traditional purification of tagged proteins, but yields 7 mg of active enzyme per liter of culture.”

2. “Our fast, single-day purification protocol results in a stable and active complex that can be used in most protein biochemistry laboratories for drug screening, as well as for routine functional and structural studies.”

-This approach of a one-day prep shows the protein is active, but they did not show the prep to be suitable for structural studies. Please remove that claim. 

This sentence has been modified as follows: “Our fast, single-day purification protocol results in a stable and active complex that can be used in most protein biochemistry laboratories for drug screening as well as for functional studies.”

3. “Depending on the applications, the N-terminal 14his-tag fused to nsp8 can be easily

removed following TEV-protease cleavage.”

-This statement should be modified to state that they have not tested if the tag is removable and the effect on the prep. If the authors did remove the tag, they should show the resulting gels and biochemical activity. 

This sentence has been modified as follows: “Depending on the applications, the N-terminal 14his-tag fused to nsp8 can be removed following TEV-protease cleavage. We recommend to do so after the HiTrapQ step. The RdRp could thus be incubated overnight with TEV-protease at 4˚C in the presence of 1mM DTT before the final size exclusion chromatography. However, since the 14his tag is cleavable, we did not verify RdRp activity after cleavage in the present study.”

4. The plasmid is not yet deposited on Addgene- do the authors plan to do so when the paper is published? If so, change “Our construction has been made available to the entire

scientific community through the Addgene plasmid repository (Addgene ID: 165451) upon publication of this manuscript.”

The plasmid is indeed not yet available and still held for publication. The sentence has been modified as suggested by the reviewer.

5. In the conclusion: 

“The resulting sample is pure, homogeneous, properly active, and therefore suitable for drug screening and extensive site directed mutagenesis, as well as for functional and structural studies.”

 -Reword as pure and homogeneous means the same in this context. Properly active is redundant; change to active. Remove extensive. To state that the resulting sample (from the single-day prep) is suitable for structural studies has not been shown. They could say that the one-day prep is ideal for biochemical studies, and while the three-step prep SHOULD be suitable for structural studies BUT that remains to be shown. Otherwise, the conclusion can be construed as misleading. 

The conclusion has been modified as follows: “Motivated by E. coli’s broad accessibility, ease of culture, rapid growth rates and proven scalability, we developed an efficient expression and purification system for the SARS-CoV-2 RdRp complex in this bacterial host. Active RdRp can be immobilized and isolated from bacterial lysate in one step by nickel affinity purification, facilitating high-throughput screens and biochemical studies. Furthermore, the three-step purification protocol yields an intact and correctly assembled (nsp7)(nsp8)2(nsp12) complex that should be suitable for other purpose as structural studies. Our construction has been made available to the entire scientific community through the Addgene plasmid repository (Addgene ID: 165451) upon publication of this manuscript.”

---

## [Editor Report · Decision Letter 2]

12 Apr 2021

Fast and efficient purification of SARS-CoV-2 RNA dependent RNA polymerase complex expressed in Escherichia coli

PONE-D-20-24456R2

Dear Dr. Sauguet,

We’re pleased to inform you that your manuscript has been judged scientifically suitable for publication and will be formally accepted for publication once it meets all outstanding technical requirements.

Kind regards,

Filippo Prischi

Academic Editor

PLOS ONE
---

## [Editor Report · Acceptance letter]

15 Apr 2021

PONE-D-20-24456R2 

Fast and efficient purification of SARS-CoV-2 RNA dependent RNA polymerase complex expressed in Escherichia coli 

Dear Dr. Sauguet:

I'm pleased to inform you that your manuscript has been deemed suitable for publication in PLOS ONE. Congratulations! Your manuscript is now with our production department. 

Kind regards, 

on behalf of

Dr. Filippo Prischi 

Academic Editor

PLOS ONE